# Cancer Stem-Like Phenotype of Mitochondria Dysfunctional Hep3B Hepatocellular Carcinoma Cell Line

**DOI:** 10.3390/cells10071608

**Published:** 2021-06-27

**Authors:** Yu-Seon Han, Eui-Yeun Yi, Myeong-Eun Jegal, Yung-Jin Kim

**Affiliations:** 1Department of Molecular Biology, Pusan National University, Busan 46241, Korea; yshan1116@pusan.ac.kr (Y.-S.H.); euiyeunyi@pusan.ac.kr (E.-Y.Y.); youll711@pusan.ac.kr (M.-E.J.); 2Korea Nanobiotechnology Center, Pusan National University, Busan 46241, Korea

**Keywords:** mitochondria, mitochondrial dysfunction, cancer stem-like cell, self-renewal, chemotherapy resistance, angiogenesis

## Abstract

Mitochondria are major organelles that play various roles in cells, and mitochondrial dysfunction is the main cause of numerous diseases. Mitochondrial dysfunction also occurs in many cancer cells, and these changes are known to affect malignancy. The mitochondria of normal embryonic stem cells (ESCs) exist in an undifferentiated state and do not function properly. We hypothesized that mitochondrial dysfunction in cancer cells caused by the depletion of mitochondrial DNA might be similar to the mitochondrial state of ESCs. We generated mitochondria dysfunctional (ρ^0^) cells from the Hep3B hepatocellular carcinoma cell line and tested whether these ρ^0^ cells show cancer stem-like properties, such as self-renewal, chemotherapy resistance, and angiogenesis. Compared with Hep3B cells, the characteristics of each cancer stem-like cell were increased in Hep3B/ρ^0^ cells. The Hep3B/ρ^0^ cells formed a continuous and large sphere from a single cell. Additionally, the Hep3B/ρ^0^ cells showed resistance to the anticancer drug doxorubicin because of the increased expression of ATP-binding cassette Subfamily B Member 1. The Hep3B/ρ^0^ conditioned medium induced more and thicker blood vessels and increased the mobility and invasiveness of the blood vessel cells. Therefore, our data suggest that mitochondrial dysfunction can transform cancer cells into cancer stem-like cells.

## 1. Introduction

Mitochondria are cellular powerplants that produce the energy required for basic cellular functions. In addition to energy supply, mitochondria are organelles that maintain other important processes, such as apoptosis and calcium buffering [1,2,3,4]. Mitochondria remain in a functional state throughout the cell cycle, but when dysfunction occurs, they are involved in many diseases, including cardiovascular disease, diabetes, and neurodegenerative diseases [4,5,6,7,8,9,10,11,12]. Mitochondrial dysfunction is observed in many cancers and promotes the progression to malignancy [13,14].

Somatic cells contain efficient mitochondria and accumulate active mitochondria to supply energetic demands. However, mitochondria in embryonic stem cells (ESCs) are different. ESCs are self-renewing pluripotent cells. Mitochondria in oocytes are vigorous, but shortly after fertilization, the mitochondria are diluted by cell division and become much less active [15,16]. Under hypoxic conditions, there is a metabolic change from oxidative phosphorylation (OXPHOS) to glycolysis. ESCs gain energy through glycolysis, similar to cancer cells because they are in a hypoxic environment where blood supply is limited. The mitochondria of ESCs are characterized by an immature cristae structure with perinuclear localization, a small number of mitochondrial DNA (mtDNA) copies, and reduced mass [15,16,17,18].

Mitochondrial function and energy metabolism play an important role in the acquisition and maintenance of cancer stem-like cell (CSC) properties, and CSCs contain a low quantity of mitochondria and low reactive oxygen species (ROS) levels [19,20]. CSCs are mainly considered as glycolysis-dependent cells but also utilize OXPHOS depending on their environment and location. In addition to mitochondrial metabolism, mitochondrial biogenesis, mitophagy, and oxidative stress are also involved in CSC regulation [21,22].

CSCs are a subpopulation of cancer cells with characteristics similar to those of normal stem cells. CSCs can be identified by characteristics, including the ability to form new tumors from a small number of cells, self-renewal potential, possession of stem cell markers, therapy resistance, metastasis, and angiogenesis [23]. A common isolation method for CSCs is using the unique and characteristic biomarkers and self-renewal capability of CSCs. It is known that the major pathways involved in CSC self-renewal include the TGF-β, Notch, and Wnt pathways [24,25]. Chemotherapy resistance is another characteristic of CSCs, attributed to the high expression level of the ATP-binding cassette (ABC) transporter, a membrane protein that acts as a drug efflux pump [26]. Angiogenesis plays an important role in maintaining the CSC niche and is regulated by growth factors, such as VEGF and EGF [27].

Various causes of mitochondrial dysfunction in cancer cells are known, such as mtDNA mutations and enzyme defects [13,14]. Mitochondrial dysfunction in cancer cells can be induced by gene editing or by depleting mtDNA using chemicals [28,29,30]. In a previous study, we generated mitochondria dysfunctional (ρ^0^) cells, Hep3B/ρ^0^ cells, by mtDNA depletion and showed that these Hep3B/ρ^0^ cells exhibited an epithelial-mesenchymal transition, a malignant phenotype. We also demonstrated a collapsed cristae structure, depleted expression of mitochondrial mRNA, and reduced oxygen consumption in Hep3B/ρ^0^ cells [31]. Therefore, we assumed that these mitochondrial changes in Hep3B/ρ^0^ cells could be similar to the mitochondrial state of ESCs. Mitochondrial changes have also been observed in induced pluripotent stem cells (iPSCs). Mitochondria in iPSCs revert to an undifferentiated ESC-like state characterized by an immature cristae structure and perinuclear distribution, decreased mtDNA content, and decreased ROS production [32,33]. Thus, in this study, we investigated whether mitochondria dysfunctional cells, Hep3B/ρ^0^ cells, exhibit CSC-like properties.

## 2. Materials and Methods

### 2.1. Materials

Dulbecco’s modified Eagle’s medium (DMEM), 2× RPMI 1640 medium, M199 medium, fetal bovine serum (FBS), sodium pyruvate (100 mM), trypsin-EDTA solution, and phosphate-buffered saline were purchased from Welgen, Inc. (Daegu, Korea). Uridine was purchased from Sigma-Aldrich (St. Louis, MO, USA). Basic fibroblast growth factor (bFGF) and heparin were obtained from PeproTech, Inc. (Rocky Hill, NJ, USA). Antibiotic-antimycotic solution and penicillin-streptomycin were purchased from Cytiva (Marlborough, MA, USA). Matrigel and 8-μm pore Transwell filter chambers were purchased from Corning, Inc. (Tewksbury, MA, USA). Qiazol was purchased from Qiagen (Hilden, Germany). All oligo primers were generated by Cosmo Genetech, Co., Ltd. (Seoul, Korea). The antibodies were purchased from the following suppliers: Cell Signaling Technology Inc. (Danvers, MA, USA) and Santa Cruz Biotechnology, Inc. (Dallas, TX, USA). SB-431542 was purchased from Tocris Bioscience (Ellisville, MO, USA). Doxorubicin hydrochloride and ABCB1-specific inhibitor R(+)-verapamil monohydrochloride hydrate were purchased from Sigma-Aldrich (St. Louis, MO, USA). Rhodamine 123 and agarose were purchased from Invitrogen (Eugene, OR, USA).

### 2.2. Cell Culture

Human hepatocellular carcinoma cells (Hep3B) were obtained from the Korean Cell Line Bank (Seoul, Korea). Hep3B cells were grown in high-glucose DMEM supplemented with 10% FBS and 1% antibiotic-antimycotic solution in a 37 °C incubator with a humidified atmosphere containing 5% CO_2_. The generation of Hep3B/ρ^0^ has been previously reported [31]. Hep3B/ρ^0^ cells were grown in high-glucose DMEM supplemented with 10% FBS, 50 μg/mL uridine, 100 mM sodium pyruvate, and 1% antibiotic-antimycotic solution in a 37 °C incubator with a humidified atmosphere containing 5% CO_2_. Human umbilical vein endothelial cells (HUVECs) were obtained from ATCC (Manassas, VA, USA). HUVECs were grown in M199 supplemented with 20% FBS, 20 ng/mL bFGF, 100 units/mL penicillin, and 100 μg/mL streptomycin in a 37 °C incubator with a humidified atmosphere containing 5% CO_2_.

### 2.3. Real-Time PCR

Total RNA from Hep3B and Hep3B/ρ^0^ cells was isolated using Qiazol reagent according to the manufacturer’s instructions. First-strand cDNA was synthesized using M-MLV reverse transcriptase (Elpis, Deagu, Korea) according to the manufacturer’s instructions. After the first-strand cDNA was synthesized, real-time PCR was performed. The data were analyzed using the Opticon Monitor Quantitation program.

### 2.4. Sphere Formation Assay

The sphere-forming capability of the cells was determined using a 96-well plate with a non-adhesive surface. For the sphere-forming culture, the bottom of the plate was pre-coated with 0.4% agarose. A single cell was seeded into each well. The total medium volume of each well was set to 200 μL. The spheres were incubated for 24 d to examine growth capability.

The primary spheres were incubated for 14 d. Then, primary spheres were resuspended as single cells, and one cell per well was seeded on a pre-agarose-coated 96-well plate. Secondary spheres were allowed to form for 14 d. Sphere formation in the third passage was verified in the same manner as described above. The interval for each passage of sphere formation was 14 d. Sphere formation was observed and imaged using a microscope.

For cancer stem-like phenotype, cells were seeded at one cell/well in an agarose-coated 96-well plate. Cells were incubated and treated with the TGF-β inhibitor SB-431542 for 14 d. On Day 14 of primary sphere culture, primary spheres were resuspended as single cells and one cell per well was seeded in an agarose-coated 96-well plate with SB-431542 as the medium.

Drug resistance was examined using a 3D culture system. Cells were seeded at 1000 cells/well in an agarose-coated 96-well plate. The cells were allowed to form spheres for 4 d. On Day 4, the mice were treated with doxorubicin and verapamil for 7 d.

### 2.5. Colony Formation Assay

A 0.8% agarose gel was prepared in advance. The 0.8% agarose gel and 2× RPMI 1640 media were mixed at a ratio of 1:1. The final concentration of agarose was 0.4%, and the 0.4% agarose with 2× RPMI 1640 medium was maintained at 50 °C. The bottom of the 35-mm dishes was coated with 0.4% agarose in 2× RPMI 1640 medium. Hep3B/ρ^0^ cells were seeded at a density of 1 × 10^4^ cells/dish in 35 mm dishes. After 4 weeks, the colony formation ability was assessed by counting the number of colonies (in three sections with a 20 mm diameter) using a phase-contrast microscope. Representative images were photographed. Hep3B cells were plated at the same density as that of the control.

### 2.6. Preparation of CM

Hep3B and Hep3B/ρ^0^ cells were seeded at 2 × 10^6^ cells/dish in 100 mm dishes in 10% FBS-containing medium and allowed to adhere for 1 d. Then, the cells were washed with phosphate buffered saline, and the medium was changed in 1% serum-free DMEM. After cultivation, the conditioned medium (CM) was harvested 24 h later and centrifuged at 1000 rpm for 1 min to avoid cell debris. The collected CM was centrifuged at 5000 rpm for 5 min at 4 °C and filtered through a 0.22 μm filter (Sartorius, Goettingen, Germany). For in vitro angiogenesis studies, Hep3B and Hep3B/ρ^0^ CMs were diluted with serum-free DMEM. All CMs used in the experiment were freshly prepared, stored at −80 °C, and used in one vial for each experiment.

### 2.7. In Vitro Wound Migration Assay

HUVECs were plated on 24-well plates (SPL, Gyeonggido, Korea) at 2 × 10^5^ cells and incubated overnight. The cells were scratched using a P200 pipette tip. After wounding, the cultures were further incubated in media supplemented with 1% FBS and CM. Cells were allowed to migrate for 24 h. Migration patterns were observed using a phase-contrast microscope and photographed. The wound diameters were photographed at 18–24 h. Wound closure was determined by optical microscopy at 40× magnification. Migration was quantified by counting the number of cells that moved beyond the reference line.

### 2.8. In Vitro Transwell Invasion Assay

The invasion capacity of the cells was determined using a 24-well Transwell system. The upper side of the Transwell membrane was coated with 1 mg/mL Matrigel in 10 μL/well. The cells were seeded at a density of 2 × 10^4^ cells in 100 μL of serum-free media with CM to the upper compartment of the Transwell and the full medium was added to the lower side. The cells were incubated for 24 h at 37 °C in 5% CO_2_. Cells were incubated at 37 °C for 24 h, fixed with methanol, and stained with hematoxylin and eosin. Cells on the upper surface of the membrane were removed by wiping with a cotton swab. Cell invasion was determined by counting whole cell numbers in a single filter by optical microscopy at 40× magnification. Each sample was assayed in duplicate, and independent experiments were repeated three times.

### 2.9. In Vitro Tube Formation Assay

HUVECs (2 × 10^4^ cells) were seeded on a layer of previously polymerized Matrigel and treated with or without CM and SB-431542. The Matrigel culture was incubated at 37 ℃. After 24 h, changes in cell morphology were captured using a phase-contrast microscope and photographed at 40× magnification. Each sample was assayed in duplicate, and independent experiments were repeated three times.

### 2.10. Crystal Violet Assay

The crystal violet assay was based on DNA staining. Cells were seeded at 5 × 10^3^ cells/well in a 24-well plate and incubated overnight. The cells were further incubated in media containing doxorubicin (2 μM) and verapamil (5 μM) for 72 h. Next, 1 mg/mL crystal violet reagent was added directly to each well. After 10 min, the reagent was removed and the cells were washed. The stained cells were dissolved in 10% sodium dodecyl sulfate (SDS). The absorbance of the crystal violet solution was determined by measuring the absorbance at 570 nm. Each sample was assayed in triplicate and the experiment was repeated three times.

### 2.11. Western Blot Analysis

Total cell lysates were prepared using PRO-PREP protein extraction solution (iNtRON Biotechnology, Korea), including 1 mM sodium orthovanadate, an inhibitor of serine/threonine protein phosphatase. Equal amounts (30 μg) of samples were resolved by electrophoresis on 10% SDS-polyacrylamide gels, transferred to a PVDF membrane, and sequentially probed with appropriate antibodies. Primary antibodies were used at a dilution of 1:1000 in 1×blocking solution (Biofact, Daejeon, Korea). This signal was developed using an enhanced chemiluminescence (ECL) detection system (Advansta, San Jose, CA, USA).

### 2.12. Terminal Deoxynucleotidyl Transferase Biotin-dUTP Nick and Labeling (TUNEL) Assay

The TUNEL assay was performed using the In Situ Cell Death Detection Kit following the manufacturer’s instructions (Roche, Switzerland). Cells were seeded on coverslips and fixed with freshly prepared paraformaldehyde for 1 h at room temperature. The coverslips were then washed with Hank’s balanced salt solution and incubated in permeabilization solution for 5 min. Then, 50 μL of TUNEL reaction mixture was added on coverslips, followed by incubation in a humidified chamber for 1 h at 37 °C in the dark. Finally, the cells were mounted. Cells were examined with an iRiS™ Digital Cell Imaging System (Logos Biosystems, Gyeonggi-do, Korea) using a 40× lens.

### 2.13. Rhodamine 123 Retention Assay

Hep3B and Hep3B/ρ^0^ cells were seeded at a density of 1 × 10^6^ cells in a 100-mm dish and incubated for 24 h. They were then treated with 2 μM rhodamine 123 for 1 h. Cells were collected and washed with HBSS, resuspended in 1 mL HBSS at a density of 1 × 10^6^ cells/mL, and then analyzed with a flow cytometer (FC500, Beckman Coulter, Fullerton, CA, USA). The fluorescence of rhodamine 123 was measured at 525 nm.

### 2.14. Statistical Analysis

Data are presented as the means ± standard deviation (SD). Statistical comparisons between groups were performed using the two-tailed *t*-test. * *p* < 0.05, ** *p* < 0.005, *** *p* < 0.0005, **** *p* < 0.0001 was considered statistically significant, and statistical analyses were performed using Prism 9.

## 3. Results

### 3.1. Mitochondria Dysfunctional (ρ^0^) Cells Acquire Self-Renewal Potential

In our previous report, we showed that mitochondrial cristae structure was damaged in Hep3B/ρ^0^ cells [31]. This structure resembled the immature mitochondria in ESCs [15,16,17,18]. We examined the cancer stem-like properties of Hep3B/ρ^0^ cells.

Stemness-associated genes, such as Oct4, Nanog, and Sox2, have been reported to regulate CSC properties [24,25]. First, the expression of ESC markers Oct4, Sox2, Nanog, and Klf4, was evaluated by qRT-PCR. Our findings indicated an increase in Oct4 and Nanog expression by approximately 2-fold and Sox2 by 6-fold in Hep3B/ρ^0^ cells compared to Hep3B cells. However, Klf4 expression level was lower than half of the Hep3B expression level in Hep3B/ρ^0^ cells (Figure 1A). We also examined protein levels by Western blot analysis, which revealed elevated protein expression of the respective molecules (Figure 1B). TGF-β, Wnt5a/b, and Notch are important for maintaining cancer stem-like properties [24,25]. We found that the Hep3B/ρ^0^ cells showed higher expression levels of these compounds compared to the Hep3B cells (Figure 1C).

Anchorage-independent growth is considered a hallmark of carcinogenesis. The colony formation assay is a well-established method for assessing malignancy [34]. We performed a colony formation assay to compare the tumorigenic potential of Hep3B cells and Hep3B/ρ^0^ cells. As shown in Figure 1D, Hep3B/ρ^0^ cells formed larger and 1.5 times as many colonies than Hep3B cells in soft agar, indicating that compared to Hep3B cells, in vitro oncogenic capabilities were significantly enhanced in Hep3B/ρ^0^ cells.

Sphere formation is one way to identify and isolate CSCs, which is a better way than exploring surface markers to represent a cancer stem-like phenotype [35]. We assumed that if Hep3B/ρ^0^ cells acquired stem-like properties, those Hep3B/ρ^0^ cells would form spheres from a single cell. We placed one Hep3B/ρ^0^ cell in each well of a 96-well plate and allowed them to form spheres to test this. As shown in Figure 1E, Hep3B/ρ^0^ cells formed relatively large spheres. In contrast, Hep3B cells did not form spheres. Except for cells attached to the agar-coated plate (approximately 20%), all non-attached Hep3B/ρ^0^ cells formed spheres. The self-renewal potential of Hep3B/ρ^0^ cells was also examined using three rounds of serial passaging (Figure 1F). The sphere created for 2 weeks was separated into a single cell, and the single-cell was again able to form a new sphere (Figure 1F).

### 3.2. Hep3B/ρ^0^ Cells Exhibit Chemotherapy Resistance via ATP-Binding Cassette Subfamily B Member 1 (ABCB1)

We attempted to demonstrate chemotherapy resistance in Hep3B/ρ^0^ cells, a cancer stem-like phenotype. To determine whether there is a difference in the chemosensitivity of Hep3B and Hep3B/ρ^0^ cells, the cytotoxic effect of doxorubicin (DX) was examined. The crystal violet staining assay revealed that Hep3B/ρ^0^ cells had higher cell viability (Figure 2A). The level of apoptosis was compared between Hep3B cells and Hep3B/ρ^0^ cells after DX treatment using the TUNEL assay. TUNEL-positive cells were observed after DX treatment, with significantly more TUNEL-positive cells among the Hep3B cells (Figure 2B). These results indicate that DX treatment of Hep3B cells induces apoptosis, and Hep3B/ρ^0^ cells are DX-resistant.

ABC transporters, including ABCB1 and Subfamily G Member 2 (ABCG2), mediate drug efflux to attenuate the accumulation of chemotherapeutic drugs in cancer cells [23,26]. We detected the upregulation of ABCB1 and ABCG2 in Hep3B/ρ^0^ cells (Figure 2C). Because rhodamine 123 uses the same pathways to pass through the membrane as conventional drugs used in oncologic treatments, by measuring the optical density (OD) of this fluorescent substance [36], we were able to determine the activity of ABC transporters, especially ABCB1, directly. The peak of Hep3B/ρ^0^ cells was located on the left side of the peak of Hep3B cells (Figure 2D). This result indicated that Hep3B/ρ^0^ cells with high ABCB1 activity pumped the fluorescent substance, rhodamine 123, and had lower OD values than Hep3B cells. The upregulation of ABCB1 and ABCG2 may be responsible for the increased resistance and reduced efficacy of DX. To validate our hypothesis, verapamil (VR) was used to inhibit ABCB1 to evaluate the role of ABCB1 in maintaining chemotherapy resistance [23,26]. We found that VR treatment effectively decreased the viability of Hep3B/ρ^0^ cells (Figure 2E). To evaluate drug resistance in a 3D-culture system, sphere cells were treated with DX and cultured in ultra-low attachment surface plates (agar coating). We reconfirmed that the sphere-forming cells of Hep3B/ρ^0^ had greater tolerance to DX treatment than Hep3B cells and that co-treatment with DX and VR induced cell death (Figure 2F).

We measured the expression levels of apoptotic molecules in both cell lines. As shown in Figure 2G, activation of caspase-3 and poly (ADP-ribose) polymerase (PARP) was observed in Hep3B cells, in which apoptosis was induced by DX treatment. However, activation of caspase-3 and PARP was not observed in Hep3B/ρ^0^ cells after DX treatment. When DX and VR were combined, these molecules were activated in Hep3B/ρ^0^ cells. Taken together, these results indicate that ABCB1 plays a crucial role in chemotherapy resistance in Hep3B/ρ^0^ cells.

### 3.3. Hep3B/ρ^0^ Cells Induce Tumor Angiogenesis in HUVECs

Recently, angiogenesis has been recognized as the CSC phenotype. Angiogenesis plays a key role in tumor development and is the process by which blood vessels proliferate inside and around the tumor to supply nutrients and oxygen to malignant cells [37]. To identify the promotion of angiogenesis by Hep3B/ρ^0^ cells, we examined tube formation, mobility and invasiveness ability in CM-treated HUVECs, representing angiogenesis in vitro. The chemotactic effects of co-culture can be replaced with CM derived from the direct culture of Hep3B and Hep3B/ρ^0^ cells, suggesting that the paracrine factors released in the CM play an important role in angiogenesis.

A tube formation assay using HUVECs in vitro was performed to measure the ability of endothelial cells to form capillary-like structures. As shown in Figure 3A, HUVECs treated with Hep3B/ρ^0^ CM formed twice as many tubes and formed thick and stable tubes.

We investigated the mobility using a wound migration assay. Figure 3B shows the representative migrating state at 24 h after creating a scratch in the cell monolayer. HUVECs treated with Hep3B/ρ^0^ CM showed increased mobility and migrated faster. The number of HUVECs that moved to the scratch area was 1.6 times higher with Hep3B/ρ^0^ CM treatment than Hep3B CM treatment (Figure 3B). In the investigation of invasiveness using a Transwell system coated with Matrigel, Hep3B/ρ^0^ CM treatment increased the invasive capacity of HUVECs by approximately two times (Figure 3C).

By enhancing tube formation and the migration and invasive abilities of HUVECs treated with Hep3B/ρ^0^ CM, we speculated that mitochondrial dysfunction increased angiogenesis properties and attempted to investigate the levels of certain proteins in Hep3B/ρ^0^ CM. Because VEGF and TGF-β play critical roles in angiogenesis [25,27,37], we examined whether Hep3B/ρ^0^ CM treatment increased VEGF and TGF-β levels in HUVECs. As shown in Figure 3D, Hep3B/ρ^0^ CM treatment induced the expression of VEGF, TGF-β, KDR, and TGF-β receptors in HUVECs.

### 3.4. TGF-β Modulates Stemness, Angiogenesis in Hep3B/ρ^0^ Cells

Because TGF-β is elevated in Hep3B/ρ^0^ cells and is known to induce angiogenesis [25,27], we speculated that TGF-β would act as a key molecule for CSC phenotypes in Hep3B/ρ^0^ cells. To determine whether mitochondrial dysfunction-induced TGF-β is responsible for the in vitro CSC phenotypes, sphere formation and angiogenesis were examined in the presence of a TGF-β inhibitor, SB-431542, a small molecule ATP-mimetic inhibitor of kinase activity [38]. As expected, SB-431542 potently inhibited the self-renewal and angiogenic properties of Hep3B/ρ^0^ cells (Figure 4).

Inhibition of TGF-β did not affect the formation of the first sphere by Hep3B/ρ^0^ cells but appeared to prevent serial sphere formation (Figure 4A). We also observed that angiogenic phenotypes, such as tube formation, mobility, and invasiveness, were reduced by treatment with Hep3B/ρ^0^ CM and TGF-β inhibitor. Unlike in the previous experiments, we concentrated the CM by two times to effectively observe the degree of reduction. Co-treatment with Hep3B/ρ^0^ CM and TGF-β inhibitor reduced tube-forming capacity compared to Hep3B/ρ^0^ CM treatment (Figure 4B). With respect to mobility and invasiveness, treatment with a TGF-β inhibitor reduced its abilities by 51% and 34% (Figure 4C,D).

We also examined the molecular changes in HUVECs after treatment with CM and/or TGF-β inhibitor using Western blot assay (Figure 4E). The Hep3B/ρ^0^ CM treatment showed an increase in TGF-β-related signaling molecules, TGF-β, TGF-β receptor, and the downstream molecule Snail1 [25]. However, the TGF-β inhibitor treatment decreased the levels of the related molecules. These data suggest that TGF-β signaling is responsible for the CSC phenotypes induced by mitochondrial dysfunction in Hep3B/ρ^0^ cells.

## 4. Discussion

Little is known about the relationship between mitochondrial dysfunction and cancer stemness. Although ESCs are difficult to study because of ethical issues, recently published papers have argued that the mitochondria of stem cells are structurally and functionally different from those of differentiated cells [15,16,17,18,39]. Studies on iPSCs also argue that in these cells the mitochondrial state is reverted to that in ESCs [32].

In our previous paper, we reported that mitochondrial dysfunction because of mtDNA depletion caused mitochondrial structural and functional changes [31]. Thus, we hypothesized that mitochondrial dysfunction would turn cancer cells into CSCs. To test this possibility, we first identified stem cell markers and their self-renewal capabilities. Consistent with our hypothesis, stem cell marker expression was observed in Hep3B/ρ^0^ cells (Figure 1A–C).

Because self-renewal is the most obvious phenotypic difference between differentiated cells and stem cells, self-renewal ability is widely used to identify CSCs. CSCs could form new tumors in a small number of cells [23,25,27]. We created an extreme environment in which only one cell was cultured in a 96-well plate to prevent communication between cells and to confirm the stem-like characteristics of the cells. We hypothesized that if Hep3B/ρ^0^ cells acquire stem-like properties, self-renewal from a single cell would be possible. As expected, none of the Hep3B cells survived in the extreme environment, but the Hep3B/ρ^0^ cells were capable of continuous culture (Figure 1E,F).

Surface markers, such as CD133 and CD90, are widely used to identify CSCs [23,25,27], but the expression of these surface markers was not observed in Hep3B/ρ^0^ cells (data not shown). However, some studies suggest that the classification of CSCs using surface markers is not accurate [40,41,42]. Therefore, we focused on the acquisition of superior self-renewal capability in Hep3B/ρ^0^ cells regardless of the presence of the markers.

Stem cells that are involved in tissue regeneration without losing their properties for a long time have developed defense mechanisms to protect themselves from toxic environments [43,44]. CSCs are also resistant to anticancer drugs for the maintenance and production of tumors [25,26,27]. Chemotherapy resistance was confirmed using DX, the most widely used anticancer drug (Figure 2A,B) [23,26]. Hep3B/ρ^0^ cells exhibited resistance to DX treatment because of elevated expression and increased activity of ABCB1 (Figure 2C,D). According to several reports, mitochondria dysfunctional cancer cells are resistant to chemotherapy because cell death does not occur because of a breakdown of the apoptotic mechanisms [45,46]. However, we assumed that like stem cells, Hep3B/ρ^0^ cells exhibit resistance to chemotherapy using efflux pump defense mechanisms. Consistent with our assumption, although DX alone did not induce apoptosis, co-treatment with the ABCB1 inhibitor and DX resulted in cell death in Hep3B/ρ^0^ cells (Figure 2E,G). These results were confirmed using a 3D-culture system (Figure 2F).

CSCs have angiogenic properties that provide cancer growth and maintenance and the formation of a cancer stem cell niche [37]. We assumed that Hep3B/ρ^0^ cells would have angiogenic properties, and angiogenesis properties were examined using HUVECs (Figure 4). We found that Hep3B/ρ^0^ CM treatment increased tube formation ability, mobility, and invasiveness, which are the main phenotypes of angiogenesis. The expression of TGF-β and VEGF, known to induce angiogenesis phenotypes [37], was also increased by Hep3B/ρ^0^ CM treatment.

TGF-β is known as a CSC marker and modulator [47,48], so an increase in the levels of TGF-β is expected to play a major role in acquiring and maintaining cancer stem-like properties in Hep3B/ρ^0^ cells. Therefore, we investigated whether a TGF-β inhibitor would abolish cancer stem-like properties, such as self-renewal and angiogenesis in Hep3B/ρ^0^ cells (Figure 4). As expected, treatment with a TGF-β inhibitor attenuated the self-renewal and angiogenic properties of Hep3B/ρ^0^ cells.

We hypothesized that mitochondrial dysfunction induces changes in the overall state of the cell, resulting in cancer stem-like characteristics. Depletion of mtDNA blocks not only mitochondrial metabolism but also changes mitochondrial and cellular structure. Structural changes are expected to affect the signaling pathways of cells, leading to cancer stem-like phenotypes. Currently, we are generating mitochondrially dysfunctional cells from various cancer cells. In addition to Hep3B/ρ^0^ cells, other established ρ^0^ cells also exhibit cancer stem-like phenotypes. We also observed common structural changes among the ρ^0^ cells generated, and we think these structural changes might be involved in inducing cancer stem-like phenotypes. Further study could reveal the possible relationship between structural changes and cancer stem-like phenotypes in ρ^0^ cells.

## Figures and Tables

**Figure 1 cells-10-01608-f001:**
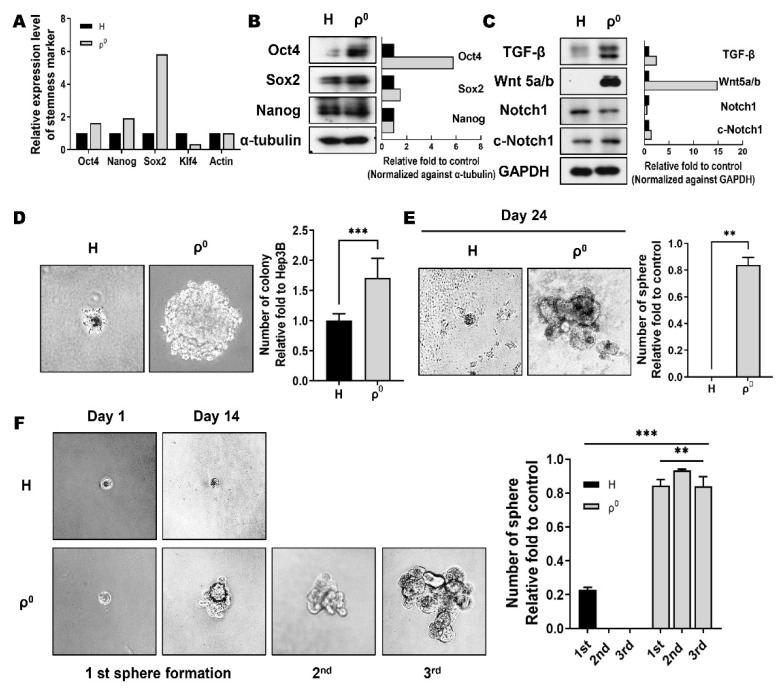
Mitochondria dysfunctional (ρ^0^) cells exhibit self-renewal properties. (**A**) Expression of embryonic stem cell (ESC)-related transcription factors was analyzed by real-time PCR and (**B**) Western blotting. Graph showing respective molecule levels normalized to the loading control. Black box: Hep3B. Grey box: Hep3B/ρ^0^. (**C**) The expression of self-renewal-related proteins was detected by Western blot analysis. Graph showing respective molecule levels normalized to the loading control. Black box: Hep3B. Grey box: Hep3B/ρ^0^. (**D**) Colony formation assay. Hep3B (**H**) and Hep3B/ρ^0^ (**ρ^0^**) cells were incubated for 4 weeks in agarose-coated 35 mm dishes (*** *p* < 0.0005). (**E**) Analysis of sphere formation capability of Hep3B and Hep3B/ρ^0^ cells. Cells were incubated for 24 d in anchorage-independent conditions (** *p* < 0.0005). (**F**) Self-renewal property was also examined through serial sphere forming in Hep3B and Hep3B/ρ^0^ cells (** *p* < 0.005, *** *p* < 0.0005).

**Figure 2 cells-10-01608-f002:**
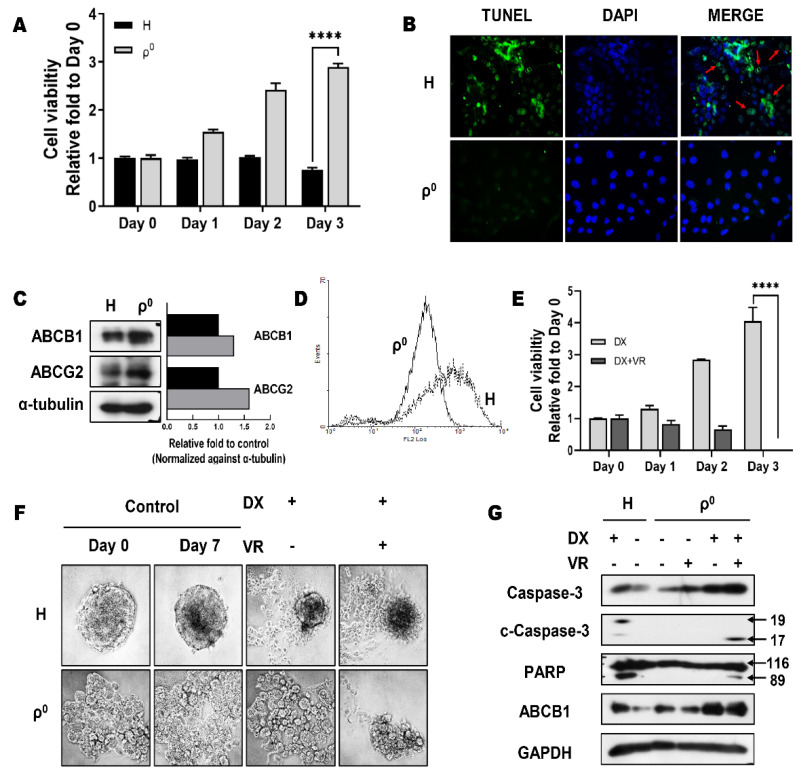
Hep3B/ρ^0^ cells exhibit drug resistance via ATP-binding cassette (ABC) transporters. The effect of mitochondrial dysfunction on drug resistance was determined using crystal violet (**A**) and TUNEL assays (**B**). (**A**) Cells were treated with doxorubicin (2 μM) for 72 h (**** *p* < 0.0001). (**B**) TUNEL staining was evaluated using fluorescence microscopy at 20× magnification. Green (arrow) indicates TUNEL-stained cells (apoptotic cells). (**C**) Western blot analysis was used to measure the protein levels of ABC transporters in Hep3B/ρ^0^ cells. Graph showing respective molecule levels normalized to the loading control. Black box: Hep3B. Grey box: Hep3B/ρ^0^. (**D**) Flow cytometry analysis of Hep3B (dashed line) and Hep3B/ρ^0^ (bold solid line) treated with rhodamine 123. Rhodamine 123 fluorescence was measured by fluorescence-activated cell sorting. (**E**) The effect of the ABCB1-specific inhibitor verapamil on doxorubicin-induced cytotoxicity was examined by MTT assay (**** *p* < 0.0001). (**F**) The cytotoxicity of doxorubicin in 3D-culture (sphere-forming). (**G**) Western blot analysis of apoptosis markers after treatment with doxorubicin and verapamil.

**Figure 3 cells-10-01608-f003:**
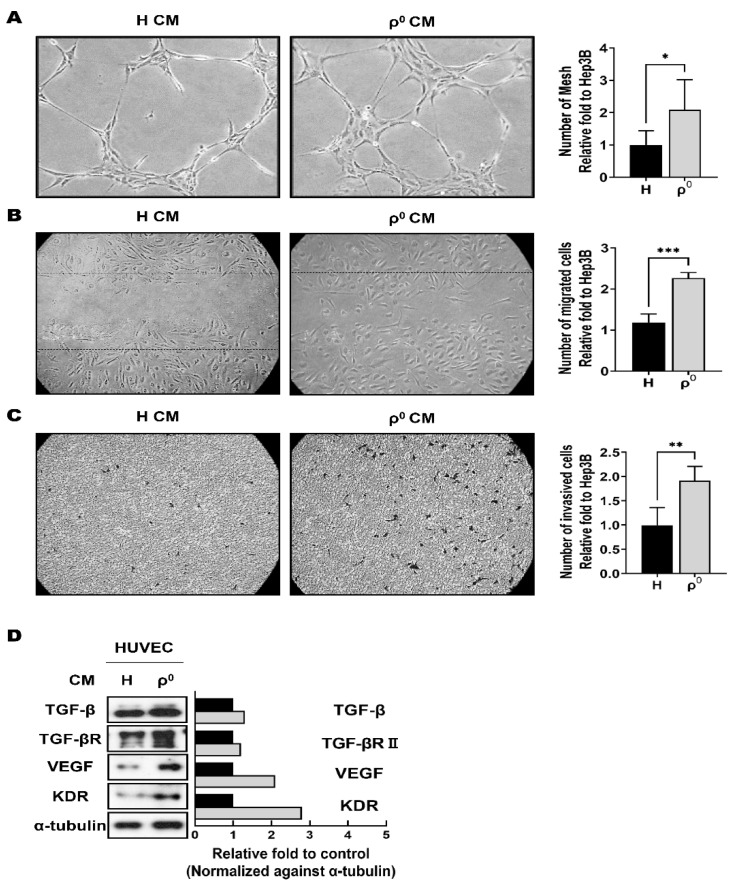
Hep3B/ρ^0^ conditioned medium (CM) promotes angiogenic effects. Angiogenic effects of CM were evaluated with tube formation (**A**), wound healing assay (**B**), and invasion assay (**C**) using human umbilical vein endothelial cells (HUVECs). (**A**) HUVECs were treated with CM for 24 h and the number of mesh was calculated (right graph) (* *p* < 0.05). (**B**) Wound healing assay for HUVEC migration using Hep3B/ρ^0^ CM (*** *p* < 0.0005). (**C**) Invasion assay for HUVEC invasion using Hep3B/ρ^0^ CM. Representative image of treated CM for 24 h and graph of the migrated cell count at 24 h. (** *p* < 0.005). (**D**) Western blot analysis of angiogenesis markers after treatment with CM in HUVECs. Graph showing respective molecule levels normalized to the loading control. Black box: Hep3B. Grey box: Hep3B/ρ^0^.

**Figure 4 cells-10-01608-f004:**
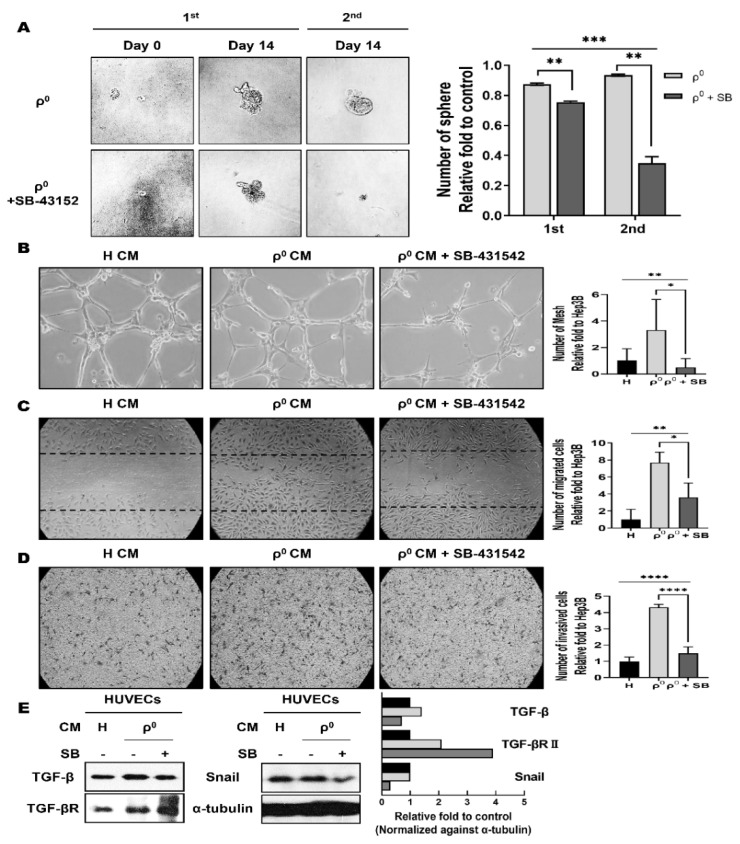
SB-431542 suppressed cancer stem-like phenotype in Hep3B/ρ^0^. SB-431542, inhibition of the TGF-β receptor, suppresses sphere formation (**A**) and angiogenic effects (**B**–**D**). (**A**) Serial sphere formation is suppressed by the inhibition of TGF-β. Representative graph of serial sphere formation after treatment with SB-431542 (** *p* < 0.005, *** *p* < 0.0005). (**B**) Suppression of tube formation following treatment with SB-431542 (* *p* < 0.05, ** *p* < 0.005). (**C**) Decreased migration ability after treatment with SB-431542 (* *p* < 0.05, ** *p* < 0.005). (**D**) SB-431542 decreased the invasive ability of HUVECs (**** *p* < 0.0001). (**E**) Western blot analysis of cancer stem-like cell-related molecules after treatment with SB-431542 and CM in HUVECs. Graph showing respective molecules levels normalized to the loading control. Black box: Hep3B CM. Grey box: Hep3B/ρ^0^ CM. Dark grey box: Hep3B/ρ^0^ CM + SB.

## Data Availability

The data presented in this study are available on request from the corresponding author.

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
