# Peer review of "Cancer Stem-Like Phenotype of Mitochondria Dysfunctional Hep3B Hepatocellular Carcinoma Cell Line"

_cells, 2021, doi:10.3390/cells10071608_

Round 1
Reviewer 1 Report
In the current work, Han et al. address an important and not completely resolved issue, concerning cancer stem-like cells (CSC), and the role of mitochondria functioning in their formation and establishment. While mitochondria dis-functioning is prevalent in CSC of hematopoietic malignancies, Glioma CSC for example, rely on mitochondrial metabolism rather than on glycolytic ATP supply. Thus, the role of mitochondria in CSC is intricate and warrants further study and exploration. In the current manuscript, the authors reveal and describe a new aspect of this topic, which extends our understanding in the dedifferentiation process of malignant cells, toward CSC. The authors show that deliberate dismantling of the mitochondria of Hep3B-hepatocellular carcinoma cells, through mitochondrial DNA depletion, endows these malignant cells with CSC properties. This an interesting finding which should be of interest to the wide audience of the "Cell" journal. Thus, after addressing the comments and suggestions bellow, this work will warrant publication in the "Cells", journal.
Basic comment:
The Introduction and Discussion sections of the manuscript should gain further depth, and in both sections, the authors should relate to the diverse aspects of the mitochondrial roles, in CSC development and establishment.
Specific comments:
- Figure 1A and B- While the RNA levels of snail were also evaluated in the qRT-PCR analysis presented in Figure 1A, snail is not mentioned among the quantified RNAs detailed in the Results section (line 213). The WB analysis presented in Figure 1B should be quantified, based on the relative tubulin levels, obtained in the two compared samples. The relative protein levels should be presented by histograms. This will give the readers a better understanding of the fold increase of each protein in the Hep3B/r0 (r0), compared to the Hep3B cells.
- Figure 2C- Based on the obtained tubulin levels, the relevant WB bands should be quantified in the Hep3B and Hep3B/r0 samples, and presented as comparative histograms.
- Figure 2E- The E label, appears mistakenly twice in the figure. The second time is most probably meant to be –Figure 2G. Arrows should be added to the left panel of the corrected Figure 2G, clearly indicating the migration distances of the cleaved Caspase-3 and PARP, products.
- Figure 3D- The relative levels of TGF-b and TGF-bR , in the WB of the Hep3B and Hep3B/r0-samples, should be quantified and presented in histograms, based on the loaded tubulin levels.
- The claimed in increase in TGF-b , and Snail levels, in HUVECs treated with CM from Hep3B/r0 , cells is not convincing. The WB in the left and right panels should be quantified based on the loaded tubulin levels.
Author Response
We appreciate your advice on our paper and have revised it to reflect your request as much as possible.
Basic comment:
The Introduction and Discussion sections of the manuscript should gain further depth, and in both sections, the authors should relate to the diverse aspects of the mitochondrial roles, in CSC development and establishment.
- A. Added in introduction as suggested.
Specific comments:
- Figure 1A and B- While the RNA levels of snail were also evaluated in the qRT-PCR analysis presented in Figure 1A, snail is not mentioned among the quantified RNAs detailed in the Results section (line 213). The WB analysis presented in Figure 1B should be quantified, based on the relative tubulin levels, obtained in the two compared samples. The relative protein levels should be presented by histograms. This will give the readers a better understanding of the fold increase of each protein in the Hep3B/r0 (r0), compared to the Hep3B cells.
A. The snail in Figure 1A was removed from the figure because it was not crucial at the beginning of the paper. - Figure 2C- Based on the obtained tubulin levels, the relevant WB bands should be quantified in the Hep3B and Hep3B/r0 samples, and presented as comparative histograms.
A. Corrected as requested. - Figure 2E- The E label, appears mistakenly twice in the figure. The second time is most probably meant to be –Figure 2G. Arrows should be added to the left panel of the corrected Figure 2G, clearly indicating the migration distances of the cleaved Caspase-3 and PARP, products.
A. Corrected as requested. - Figure 3D- The relative levels of TGF-b and TGF-bR , in the WB of the Hep3B and Hep3B/r0-samples, should be quantified and presented in histograms, based on the loaded tubulin levels.
A. Corrected as requested. - The claimed in increase in TGF-b, and Snail levels, in HUVECs treated with CM from Hep3B/r0, cells is not convincing. The WB in the left and right panels should be quantified based on the loaded tubulin levels.
A. Although the expression of TGF and Snail was not dramatically increased, TGF inhibitor treatment abolished the angiogenic phenotype.
Reviewer 2 Report
Review of Ms: ‘Cancer stem-like phenotype of mitochondria dysfunctional Hep3B hepatocellular carcinoma cell line” Yu-Seon Han et al.
This manuscript describes basic research studies on mitochondrial behaviors as relate to cancer stem-cells. The authors compared biological features of normal undifferentiated embryonic stem cells (ESCs) mitochondria with hepatocellular carcinoma cell line Hep3B. Induction of mitochondria dysfunctional (ρ0) cells in Hep3B cell line demonstrated cancer stem-like features of self-renewal, chemotherapy resistance, and angiogenesis. Authors suggested that the mitochondrial dysfunction and depletion of mtDNA in hepatocellular carcinoma cell line were similar to features of embryonic stem cancer cells (ESCs), with regard to cell anchoring, sphere-formation, resistance to known anticancer drugs (doxorubicin or verapamil) in induction of apoptosis, were demonstrated by increased expression of ATP-binding of subfamily B member1 or induction of angiogenic-like vasculogenesis.
The role of mitochondria in health and diseases, particularly the induction of cancer is a highly important and neglected topic. The in vitro methods that authors used for comparing mitochondrial dysfunction of cancer stem cells with cultures hepatocellular carcinoma cell line are suitable and useful in understanding mitophagy in cancer. Below are summary of my questions and recommendations to improve the articles:
- Major Points:
- Experiments on comparison of “Self- renewal experiments of Hep3 B/ρ0 cells…” and embryonic stem cells (ESC) in Figures 1 and 2 show sphere formation and mtDNA depletion. It would be interesting if the authors could demonstrate comparison of mitochondrial behaviors with normal hepatocytes under similar experimental conditions. Authors are recommended to prepare a table showing such the effects of available data on several factors (TGF-b) or inhibitors on mtDNA or sphere formation.
- Could these studies be comparable or complementary to those in vitro studies where uncouplers (dinitrophenol, Ouabain) are used in tumors to show ATPase activities or cell death? Authors may describe such studies in introduction or discussion of their article for comprehension of the topic (Chang et al, 2019)
- Cancer cells are defective cells that survive under hypoxic conditions as the results of mitochondrial dysfunction. Studies suggest that non-functional mitochondria and the absence of apoptotic properties of immune system during fetal growth (under hypoxic condition) are required for vasculogenesis and organogenesis. Stimuli-induced unresolved inflammation (oxidative stress) is associated with loss of highly regulated balance between apoptosis (Yin) and wound healing (Yang) properties of immunity along with mitochondrial dysfunction (mitophagy) in multistep tumor growth, angiogenesis and loss of contact inhibition (structural integrity) features of which share with fetus growth (Khatami, 2018, 2020).
Author Response
We appreciate your advice on our paper and have revised it to reflect your request as much as possible.
Manuscript ID : cells-1259758
Major Points:
- Experiments on comparison of “Self- renewal experiments of Hep3 B/ρ0 cells…” and embryonic stem cells (ESC) in Figures 1 and 2 show sphere formation and mtDNA depletion. It would be interesting if the authors could demonstrate comparison of mitochondrial behaviors with normal hepatocytes under similar experimental conditions. Authors are recommended to prepare a table showing such the effects of available data on several factors (TGF-b) or inhibitors on mtDNA or sphere formation.
- A. That opinion is interesting, but we are afraid that normal hepatocytes cannot survive our Hep3B/ρ0 generating system. There are published papers on inducing iPSCs by transforming mitochondrial metabolic processes (Spyrou J et al. Stem Cells Int. 2019) (Ishida T et al. Inflamm Regen. 2020).
- Could these studies be comparable or complementary to those in vitro studies where uncouplers (dinitrophenol, Ouabain) are used in tumors to show ATPase activities or cell death?
- A. We assume that the uncouplers would not affect Hep3B/ρ0 cell viability because of no RNA expression of mitochondrial ATPase6 and high activity of ABC transporters in the ρ0 cell.
- Authors may describe such studies in introduction or discussion of their article for comprehension of the topic (Chang et al, 2019)
- A. Added in introduction as suggested.
- Cancer cells are defective cells that survive under hypoxic conditions as the results of mitochondrial dysfunction. Studies suggest that non-functional mitochondria and the absence of apoptotic properties of immune system during fetal growth (under hypoxic condition) are required for vasculogenesis and organogenesis. Stimuli-induced unresolved inflammation (oxidative stress) is associated with loss of highly regulated balance between apoptosis (Yin) and wound healing (Yang) properties of immunity along with mitochondrial dysfunction (mitophagy) in multistep tumor growth, angiogenesis and loss of contact inhibition (structural integrity) features of which share with fetus growth (Khatami, 2018, 2020).
- A. We think that the paper took a slightly different direction from what we argued. We found papers that fit our paper and added them to the introduction.
Reviewer 3 Report
Authors demonstrated that depletion of mitochondrial DNA can transform Hep3B cancer cells into cancer stem-like cells. This article could be an interesting and robust technical note. I have some minor concerns listed below.
- Authors describe “mitochondria dysfunctional”Hep3B but indeed there were no results on the mitochondrial function assessment in the whole study.
- I believed Hep3B/ρ0 cells were successfully made but was wondering whether they remained in the ρ0 status immediately before and during each experiment.
- Prism was not properly cited (line 206)
- Some errors existed in Figure 2. For example, G was misplaced as E.
- What does the abbreviation PARP mean? (line 291)
Author Response
We appreciate your advice on our paper and have revised it to reflect your request as much as possible.
Manuscript ID : cells-1259758
- Authors describe “mitochondria dysfunctional”Hep3B but indeed there were no results on the mitochondrial function assessment in the whole study.
- A. The properties of Hep3B/ρ0 cells were described in our previous paper (Ref 31). We briefly mentioned it in the last paragraph of the introduction.
- I believed Hep3B/ρ0 cells were successfully made but was wondering whether they remained in the ρ0 status immediately before and during each experiment.
- A. We checked mtDNA expression by RT-PCR before performing experiments to confirm the Hep3B/ρ0 cell status.
- Prism was not properly cited (line 206)
- A. Corrected as requested.
- Some errors existed in Figure 2. For example, G was misplaced as E.
- A. Corrected as requested.
- What does the abbreviation PARP mean? (line 291)
- A. Corrected as requested.
Round 2
Reviewer 1 Report
The authors have adequately addressed my comments
Reviewer 2 Report
The revised Ms ID: Cell-1259758 was reviewed. The authors made some revisions demonstrated by RED lines--very little was incorporated from my recommendations, questions and suggested references. I did not see specific responses to the points that were made.
Figure legends have been improved.